# Evaluation of Saline Solutions and Organic Compounds as Displacement Fluids of Bentonite Pellets for Application in Abandonment of Offshore Wells

Waleska Rodrigues Pontes da Costa [1,*], Laura Rafaela Cavalcanti de Oliveira [1], Karine Castro Nóbrega [1], Anna Carolina Amorim Costa [1], Ruth Luna do Nascimento Gonçalves [1], Mário César de Siqueira Lima [1], Renalle Cristina Alves de Medeiros Nascimento [2], Elessandre Alves de Souza [3], Tiago Almeida de Oliveira [4], Michelli Barros [1] and Luciana Viana Amorim [1]

1 Unidade Acadêmica de Engenharia de Petróleo/Unidade Acadêmica de Estatística, Universidade Federal de Campina Grande (UFCG), Rua Aprigio Veloso, 882, Bairro Universitário, Campina Grande 58429-900, PB, Brazil; laura.rafaela@hotmail.com (L.R.C.d.O.); karine.nobrega@hotmail.com (K.C.N.); anna.amorimc@gmail.com (A.C.A.C.); ruthlunang@gmail.com (R.L.d.N.G.); mariocesar.slima@gmail.com (M.C.d.S.L.); michelli.karinne@gmail.com (M.B.); lvamorim@gmail.com (L.V.A.)
2 Unidade Acadêmica de Santo Agostinho, Universidade Federal Rural de Pernambuco (UFRPE), Rua Cento e Sessenta e Três, 300, Garapu, Cabo de Santo Agostinho 54518-430, PE, Brazil; nalenascimento@gmail.com
3 Centro de Pesquisas Leopoldo Américo Miguez de Mello (CENPES), PETROBRAS (Petróleo Brasileiro S.A.), Avenida Horacio Macedo, 250, Cidade Universitária, Rio de Janeiro 21941-915, RJ, Brazil; elessandre@petrobras.com.br
4 Departamento de Estatística, Universidade Estadual da Paraíba, Rua Baraúnas, 351, Bairro Universitário, Campina Grande 58429-500, PB, Brazil; tadolive@servidor.uepb.edu.br
* Correspondence: warodriguespc@gmail.com

**Abstract:** One of the operational challenges regarding the use of bentonite pellets as sealing materials in the abandonment of offshore fields consists of their placement inside the well. This study aimed to analyze the interaction of fluid media, consisting of saline solutions (NaCl, $CaCl_2$ and KCl) and organic compounds (diesel, glycerin and olefin), with bentonite pellets, for their applications as displacement fluids in offshore oil well abandonment operations. The physical integrity of the bentonite pellets in contact with the fluids was verified through visual inspections and dispersibility tests. Linear swelling tests were also performed to evaluate the swelling potentials of the pellets in deionized water after their contact with the fluid media. The results indicated that the NaCl, $CaCl_2$ and KCl solutions completely compromised the physical integrity of the pellets, while diesel and olefin showed the best responses regarding the structural preservation. Furthermore, the linear swelling tests showed that, even after the contact with diesel and olefin for 1 h, the bentonite pellets reached a total swelling of 78% in water after 24 h. In this way, diesel and olefin proved to be highly promising alternatives to be used as displacement fluids for bentonite pellets in wells that will be abandoned in a submarine environment.

**Keywords:** compacted clay; clay swelling; physical integrity; displacement fluid; solid transportation

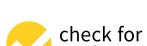



## 1. Introduction

Well abandonment operations are generally associated with the end of the productive life cycle of a field, when it becomes necessary to restore the perfect isolation between the different permeable intervals of the oil and/or gas zones and existing aquifers, preventing the migration of fluids between formations or to the surface or seabed [1]. The isolation of these zones, which may be located close to the perforations, as well as in the intermediate zones of the well or close to the surface, has been commonly performed with cement plugs [1].

The use of cement as a sealing material, although considered effective and required by most international regulatory bodies, has limitations related to the tendency of shrinking and cracking and mechanical or chemical degradation, compromising its integrity and resulting in high costs to repair the plug [2,3]. In addition, the use of this type of material requires additional costs due to the need of a work-over rig to carry out the pumping and setting of the plug, which represents a substantial plugging cost [3].

One of the alternatives suggested for the use of cement in oil well abandonment is the use of bentonite clay, which has also been widely used in the plugging of water wells, seismic shot holes and geological repositories for radioactive waste [2,4–6]. The advantages of using bentonite plugs, when compared to cement, include economic and environmental benefits and aspects related to safety and health.

The sealing properties of bentonite are related to its high swelling capacity and low permeability. These properties are ensured mainly by its physical features, such as its high cation exchange capacity, large plasticity index and large specific surface area, which are due to its mineralogy, essentially the great amount of the montmorillonite mineral [7]. In addition, the plasticity of this material makes it more reliable in the case of the formation of cracks, since it tends to heal, ensuring the integrity of the formation [1,8].

Several studies agree that hydrated bentonite is effective in isolating formations with different pressure gradients [4,9–11]. However, there are still gaps that need to be filled to ensure its effectiveness and safety, especially in offshore oil wells.

An operational challenge related to bentonite application in deep oil wells is its proper placement in the intervals defined in the abandonment project. One of the ways to ensure the formation of the bentonite plug in the appropriate location of the well, preventing premature swelling and ensuring its proper setting, is to use its compacted form, also known as pellets, since kinetic hydration is closely related to its physical conformation [1,12]. Additionally, the transportation of pellets should take place by means of a displacement fluid and should be exclusively hydromechanical, avoiding chemical interactions that could result in premature swelling or disintegration or that may impair their swelling in water once the appropriate location for the plug formation has been reached [2,13]. The mineral and chemical properties of bentonite play fundamental roles in these interactions, since its compatibility with a fluid depends on the active sites and exchangeable interlayer cations on the structure of montmorillonite, which are typically silanols and $Na^+$ and/or $Ca^{2+}$, respectively. In this case, water molecules are attracted, and bentonite swelling occurs. So, the right choice of the displacement fluid should take into account the chemical properties and compatibility with bentonite [14].

Many studies in the literature have been developed in order to characterize the swelling mechanisms of compacted bentonite based on parameters such as types of fluid for hydration, environmental conditions and physical aspects [15–18]. However, few studies analyzed the operational performance prior to the swelling stage, which is crucial, since it is related to the interaction between the pellets and the fluids that must provide their hydraulic displacements efficiently. In this sense, this study aims to analyze the interactions of fluids, consisting of saline solutions (solutions of NaCl, $CaCl_2$ and KCl at different saturations) and organic compounds (diesel, glycerine and olefin), with bentonite pellets, targeting their applications as displacement fluids in abandonment operations of offshore oil wells.

## 2. Materials and Methods

### 2.1. Materials

Compacted pellets, produced from natural bentonite by BUN—Bentonite União Nordeste (Campina Grande, Brazil), were used in this work. As displacement fluids, saline solutions and organic compounds were evaluated. The saline solutions were prepared from sodium chloride (NaCl), calcium chloride ($CaCl_2$) and potassium chloride (KCl) brines, with saturations ranging from 10 to 100%, with increments of 10% at each saturation. The amount of salt, in mg/L, used to achieve brine saturation is presented in Table 1.

**Table 1.** Amount of salt used in brines.

| Saturation | NaCl (mg/L) | CaCl₂ (mg/L) | KCl (mg/L) |
|---|---|---|---|
| 10% | 3600 | 78,600 | 3400 |
| 20% | 7200 | 157,200 | 6800 |
| 30% | 10,800 | 235,800 | 10,200 |
| 40% | 14,400 | 314,400 | 13,600 |
| 50% | 18,000 | 393,000 | 17,000 |
| 60% | 21,600 | 471,600 | 20,400 |
| 70% | 25,200 | 550,200 | 23,800 |
| 80% | 28,800 | 628,800 | 27,200 |
| 90% | 32,400 | 707,400 | 30,600 |
| 100% | 36,000 | 786,000 | 34,000 |

The organic compounds used were diesel, olefin and glycerin. The materials were provided by CENPES/PETROBRAS (Rio de Janeiro, Brazil).

*2.2. Methods*

2.2.1. Interaction under Static Conditions

The interaction of bentonite clay with each fluid was analyzed by immersing a single pellet of approximately 1.5 g in 15 mL of each fluid. The physical cohesion of the bentonite pellet was visually verified throughout the test for a maximum period of 120 min, which is the average time estimated for the displacement of pellets in a well, considering the field experience. Furthermore, in order to compare the results, its swelling in deionized water under the same conditions was evaluated.

2.2.2. Physical Integrity under Dynamic Conditions

From the previous analysis of the physical cohesion of the pellets in static conditions, dynamic tests were conducted, using compounds in which the pellets showed better maintenance of their integrity. These tests were performed based on the ISO 10416:2008 Standard [19], which specifies the methodology for shale particle disintegration test. It was adapted for pellet testing, only excluding the procedures related to milling and sieving the shale rock, since the pellets already presented adequate sizes. All other procedures, explained as follows, are in accordance with the requirements of this standard [20].

To carry out the tests, a stainless steel cell was filled with 350 mL of fluid and 20 g of bentonite pellets. The cells were exposed to shear in a Roller Oven 704 ES Fann, with a rotation of 50 rpm for 2 h. At the end of this time, the cell content (fluid and pellets) was carefully filtered through an ABNT (Associação Brasileira de Normas Técnicas, Brazilian Association of Technical Standard) number 6 sieve.

The material not retained in the sieve was considered as part of the physical disintegration of the pellet, while the material that was retained proceeded to a hot air drying oven, where it was subjected to a temperature of 60 °C for 24 h. After 24 h, the material was weighed on a high-precision analytical balance. To standardize the test conditions, 20 g of bentonite pellets without previous immersion was also placed in an oven at a temperature of 60 °C and weighed after 24 h. The disintegration of the sample was calculated according to Equation (1), specified in the ISO 10416:2008 [19].

$$D = \frac{(m_i - m_r) \times 100}{m_i} \tag{1}$$

where $D$ is the dispersibility or disintegration rate (%), $m_i$ is the initial sample mass (g) and $m_r$ is the mass of pellets retained on the sieve after drying (g).

As the material retained on the sieve was also subjected to moisture loss while drying, the lost moisture content of a sample of 20 g of bentonite pellets with no prior contact to any fluid, and after drying in the same conditions (60 °C for 24 h), was also calculated, according to Equation (2).

$$MC = \frac{(m_i - m_f) \times 100\%}{m_i} \quad (2)$$

where $MC$ is the lost moisture content, $m_f$ is the mass of the sample after drying and $m_i$ is the mass of the sample prior to drying (20 g).

In addition, specific measurements of the interaction between the pellets and the liquids were carried out, with the pellets being immersed in the media for 2 h before submitting them to the dispersibility test, excluding the loss of mass resulting from the disintegration of the pellets.

All of the tests were performed in triplicate, and the results represent the average of the measurements.

### 2.2.3. Linear Swelling

The swelling of the pellets in deionized water was evaluated after contact with the fluids, in which the physical preservation of the pellets was observed under dynamic conditions. Furthermore, in order to compare the results, their swelling in deionized water, without previous contact with such fluids, was also evaluated.

For this, linear swelling tests were carried out using a Linear Swell Meter (LSM) by Fann Instrument Company, model 2100. The tablets needed for the test were made by pressing 10 g of bentonite pellet, with a granulometry of less than 0.075 mm, corresponding to ABNT number 200 sieve at 10,000 psi (pounds per square inch) for 1 h. Linear swelling measurements were performed using an automatic digital transducer that allows for direct reading.

The test was carried out in two conditions: In the first condition, the tablets were placed in the equipment and the containers were filled with the fluid media and analyzed as displacement fluid. After one hour, the fluid medium was removed and replaced with deionized water, which remained in contact with the tablets for 24 h. The total test time was 25 h. In the second condition, the degree of swelling of the pellets in contact with deionized water for 25 h was evaluated in order to standardize the total test time. For all conditions, the swelling percentage of the tablets was calculated using the LSM 2100 software based on the values of its initial and final heights.

## 3. Results and Discussion

### 3.1. Interaction under Static Conditions

The graph in Figure 1 shows the times for which the total loss of physical integrity of the pellets, in seconds, was observed for each of the saline solutions analyzed at different saturations.

The maximum time observed for pellet disintegration was 180 s (3 min) for all saline solutions analyzed, regardless of saturation, which is much shorter than the estimated time required for their complete displacement inside the well (120 min).

The physical appearances of the bentonite pellets, before contact with any type of fluid, are shown in Figure 2. Figures 3 and 4 show the appearances observed at the end of the test for the samples that presented the shortest and longest disintegration times for each brine, respectively.

As seen in Figures 3 and 4, there was a significant physical change in the pellets that were in contact with the saline solutions, compromising their integrity. Compared with Figure 2, in which the pellets are presented before contact with the brines, the images in Figures 3 and 4 show that all brines promoted the degradation of the compacted form of bentonite, causing it to take the form of a powder or very small pieces. This may occur due to changes in the microstructure of compacted bentonite promoted by the exposure to saline solutions once the concentrated saline media promotes an osmotic suction, which

can act as an internal compaction tension, causing the particles to approach each other [21]. There is a minimization of the repulsion of the double layer that is associated with this mechanism, caused by the presence of the high cation content in the solution, resulting in an increase in the attraction between the lamellae and the consequent aggregation of the clay particles. Furthermore, the formation of aggregated structures by the clay particles becomes even more significant when the cations in the saline solution have a greater number of electrons in the valence layer, exerting a greater compaction of the double layer. This effect is observed in Figure 4.

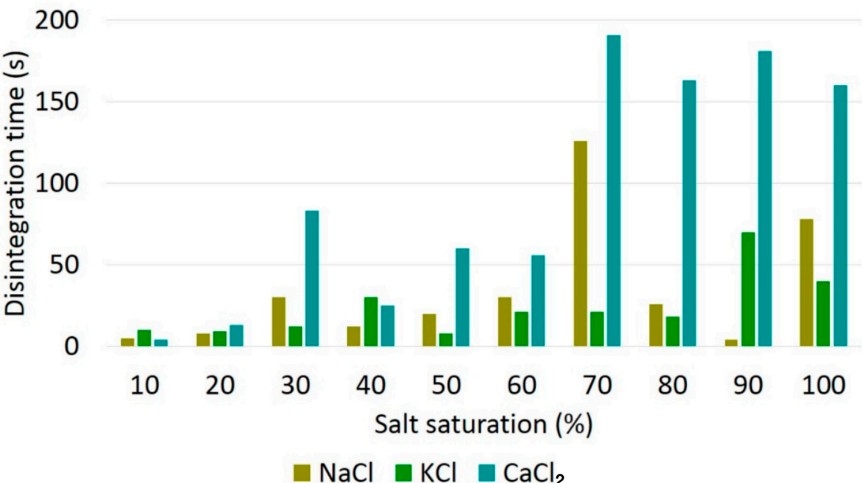

**Figure 1.** Disintegration times of pellets for saline solutions of different saturations.

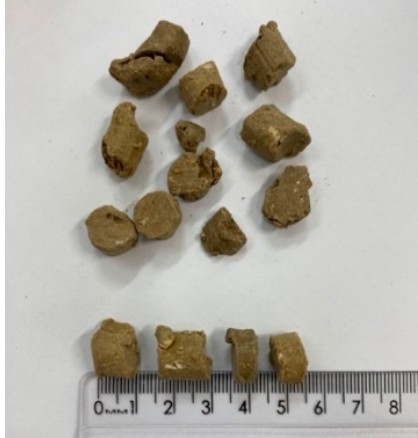

**Figure 2.** Physical aspect of bentonite pellets.

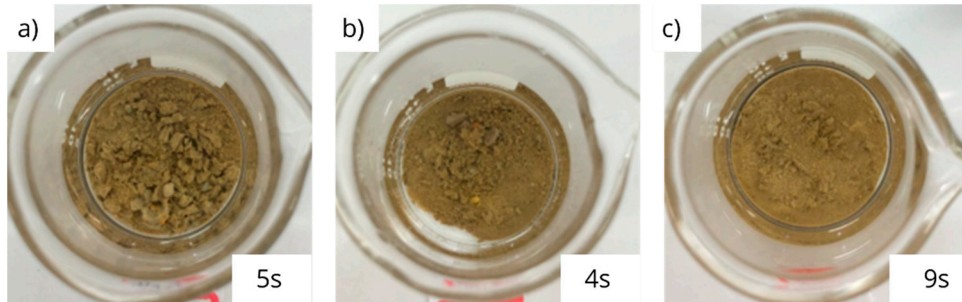

**Figure 3.** Physical appearances and disintegration times of bentonite pellets after immersion in 10% sodium chloride brine for 5 s (**a**), 10% calcium chloride brine for 4 s (**b**) and 20% potassium chloride brine for 9 s (**c**).

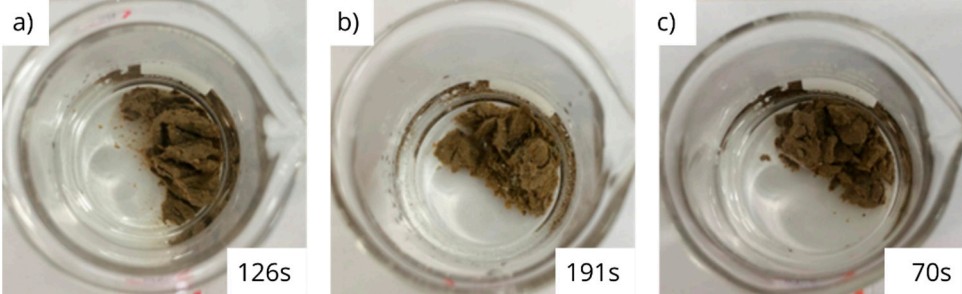

**Figure 4.** Physical appearances of bentonite pellets after immersion in 70% sodium chloride brine for 126 s (**a**), 70% calcium chloride brine for 191 s (**b**) and 90% potassium chloride brine for 70 s (**c**).

In the images presented in Figure 4, it is also observed that the pellets that were exposed to higher saline saturations presented a less expressive disintegration when compared to that presented by the pellets that were exposed to saline solutions with saturations varying between 10% and 20% (Figure 3), indicating a correlation between the mechanical properties of clay and the brine saturation. Zhang et al. (2016) [16] observed a decrease in the plastic compressibility of compacted clays with an increasing saline concentration. This behavior would also be related to the osmotic suction, which is proportional to the salt concentration, which induces the formation of aggregates and inter-aggregates in the pores, also contributing to the decrease in deformability [16,22]. The destabilization of bentonite regarding colloid sedimentation was also observed when in contact with saline solutions of NaCl and $CaCl_2$ at concentrations even lower than those used in this study [23]. It was observed that in concentrations in the order of $10^{-5}$ to $10^{-3}$ mol/L, there was a flocculation of the bentonite and, consequently, the sedimentation of its particles.

Saline solutions (especially KCl brines) have relevant applications in well engineering and can be applied in the formulation of the drilling fluids used in the drilling of clay-rich formations in order to avoid unwanted swelling [24]. However, the results show that the use of these compositions should not be extended to the abandonment of wells, since the physical integrity of the pellets is compromised when in contact with NaCl, $CaCl_2$ and KCl solutions, regardless of their concentrations.

For the evaluated organic compounds, i.e., diesel, olefin and glycerin, the immersion of the pellets resulted in the physical aspects recorded in the images presented in Figures 5, 6 and 7, respectively.

In the images presented in Figures 5 and 6, it is observed that the physical integrity of the pellets that are immersed in diesel and olefin is preserved. However, the presence of small clay fragments is observed, deposited at the bottom of the beaker, when the pellets are immersed in glycerin (Figure 7), indicating the disintegration of the pellets when immersed in this organic medium.

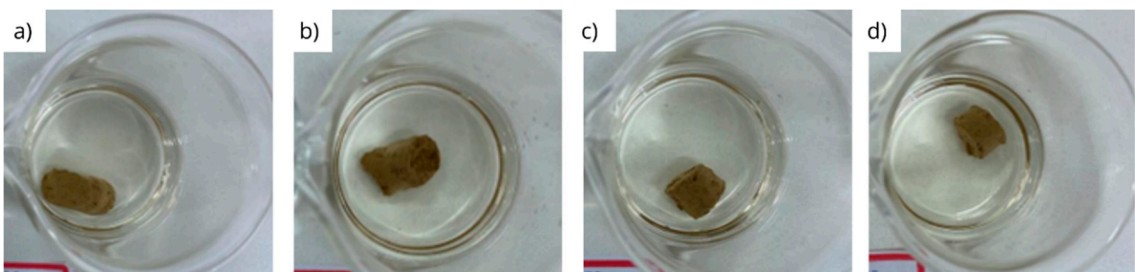

**Figure 5.** Physical appearances of bentonite pellets after immersion in diesel for 5 (**a**), 30 (**b**), 60 (**c**) and 120 (**d**) minutes.

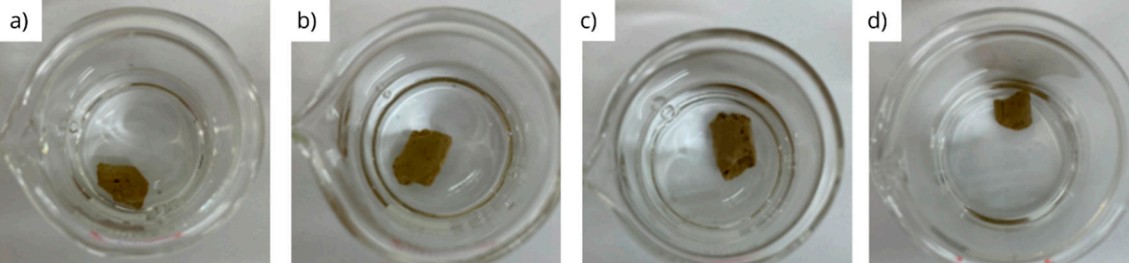

**Figure 6.** Physical appearances of bentonite pellets after immersion in olefin for 5 (**a**), 30 (**b**), 60 (**c**) and 120 (**d**) minutes.

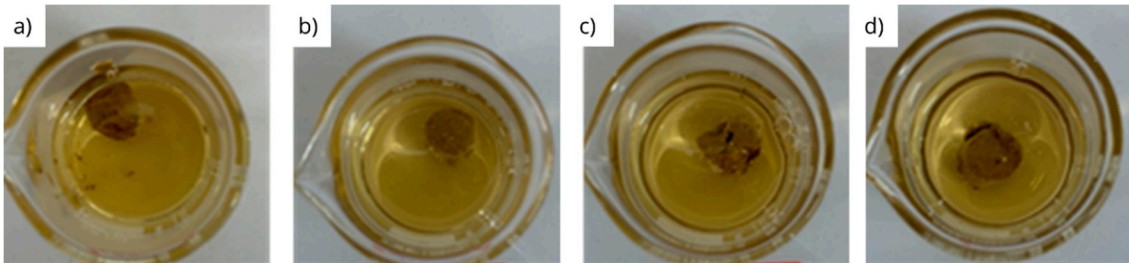

**Figure 7.** Physical appearances of bentonite pellets after immersion in glycerin for 5 (**a**), 30 (**b**), 60 (**c**) and 120 (**d**) minutes.

Natural bentonite clays have hydrophilic surfaces and, therefore, do not adsorb hydrophobic liquids, such as diesel and olefin, since these clays contain exchangeable inorganic cations that are only highly hydratable in aqueous media [25]. Therefore, the occurrence of disintegration in the presence of glycerin, although in a low proportion, may be associated with the intermolecular interaction of the clay surface with glycerol, which is the main component of this compound, since its hydrophilicity ensures the chemical affinity between them [26]. However, this chemical affinity does not constitute a possibility for the occurrence of premature swelling of the pellets in this medium since glycerin seems to have a limited invasion radius in bentonite specimens [27].

The physical appearance of a bentonite pellet after the immersion in deionized water for 60 and 120 min can be visualized in the images shown in Figure 8. It is observed that there are no significant differences in pellet swelling after 30 and 60 min, proving that almost total hydration and swelling occur in the first 60 min of contact with deionized water. This time is longer than expected for pulverized particles of bentonite, which is in the order of minutes [28].

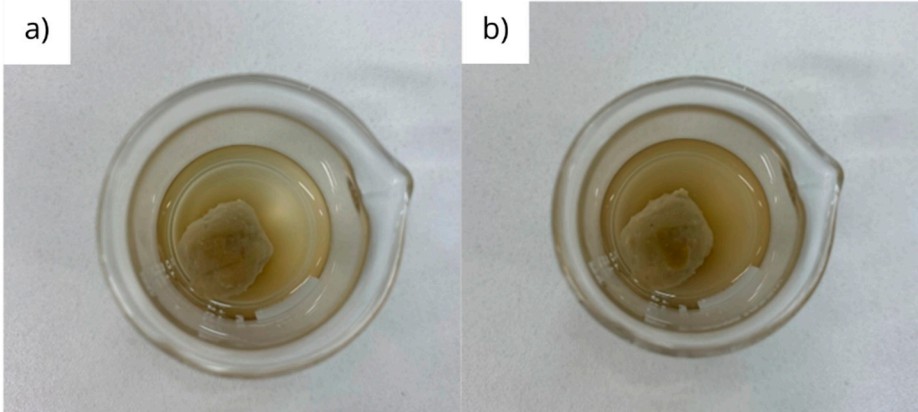

**Figure 8.** Physical aspect of bentonite pellet after immersion in deionized water for 60 (**a**) and 120 (**b**) minutes.

*3.2. Physical Integrity under Dynamic Conditions*

In view of the results obtained under static conditions, the organic compounds, diesel, glycerin and olefin, showed better tendencies to maintain the cohesion of the pellets, with total physical preservation when the organic medium was diesel or olefin, and showing partial preservation when glycerin was used. In this sense, the integrity of the pellets in contact with these fluids was also investigated through tests conducted under dynamic conditions, which are close to the turbulent conditions in which the pellets are transported into the well.

Figure 9 presents images of the organic fluid media (diesel (a), olefin (b) and glycerin (c)) filtered after the tests to evaluate the physical integrity of the pellets under dynamic conditions.

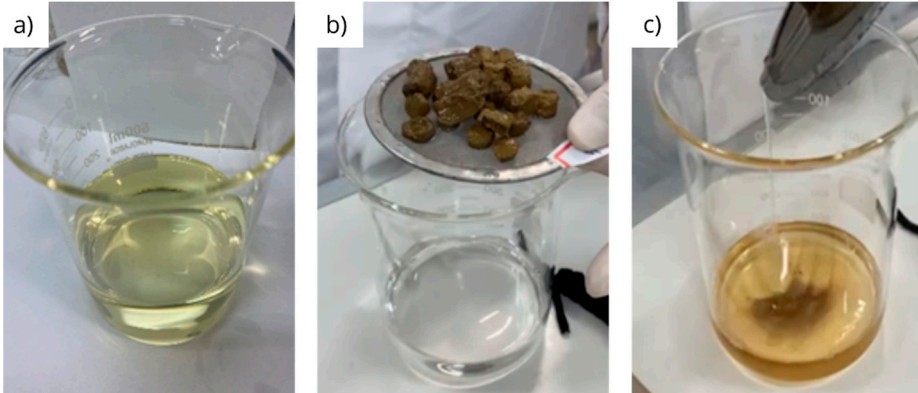

**Figure 9.** Visual appearances of diesel (**a**), olefin (**b**) and glycerin (**c**) filtered after testing under dynamic conditions.

The dynamic test significantly enhanced the disintegration of the pellets in glycerin (Figure 9c), which was also observed, in a lesser extent, under static conditions. This behavior can be clearly noted by the significant volume of fragments visualized at the bottom of the beaker containing the filtered glycerin after the test, and this is probably due to the combined action of the shear with the longer exposure time of the test when compared to the test that was carried out under static conditions. In contrast, no fragments were observed in the filtered material when diesel and olefin were used (Figure 9a,b).

Table 2 presents the initial masses of the pellets, the residual masses obtained at the end of the test (final mass) and the disintegration rates for each of the analyzed organic compounds.

**Table 2.** Disintegration rates of pellets.

| Fluid | Initial Mass (g) | Final Mass (g) | Disintegration Rate (%) |
|---|---|---|---|
| Diesel | 20.00 | 17.98 | 10.10 |
| Olefin | 20.00 | 17.49 | 12.55 |
| Glycerin | 20.00 | 17.13 | 14.35 |

The results obtained demonstrate a greater disintegration for the pellets that were immersed in glycerin, calculated at 14.35%. The disintegration rates calculated for diesel and olefin were 10.10% and 12.55%, respectively. These values were not expected, since the filtrate obtained after the tests were clear, with no visible evidence of pellet fragments. Therefore, the disintegration rate should be zero or close to it (Figure 9).

Thus, in order to understand and elucidate the results obtained, the values related to the moisture loss of the pellets and mass variation for the pellets immersed in the fluids, excluding the loss of mass resulting from the disintegration, were considered. Tables 3 and 4 present those results.

**Table 3.** Moisture loss of pellets after drying at 60 °C for 24 h.

| Initial Mass (g) | Final Mass (g) | Moisture Loss (%) |
| --- | --- | --- |
| 20.00 | 17.22 | 13.90% |

**Table 4.** Mass variations of pellets after immersion in fluid and drying at 60 °C for 24 h.

| Fluid | Initial Mass (g) | Mass after Immersion in Fluid (g) | Final Mass (g) | Mass Variation (%) |
| --- | --- | --- | --- | --- |
| Diesel | 20.00 | 20.71 | 17.86 | −10.70 |
| Olefin | 20.00 | 20.45 | 17.55 | −12.25 |
| Glycerin | 20.00 | 26.06 | 23.57 | +17.85 |

The results obtained show an average mass reduction, due to the moisture loss of the pellets without prior immersion, of 13.90% (Table 3). This value is significantly close to the disintegration rate obtained for the test carried out under dynamic conditions for the previously immersed samples in diesel and in olefin, which are 10.10% and 12.55% (Table 4). Similarly, the mass variations obtained when performing the static immersion of the pellet in diesel and olefin once again represent losses similar to those obtained in the other tests, calculated at 10.70% and 12.25%. Thus, the water that is physically attached to the edges and external surfaces of the pellets is removed during drying and considered in the calculations to obtain disintegration. As this amount of removed water does not compromise the physical cohesion of the pellets, a correction considering the moisture loss is required in order to obtain the actual rate of disintegration.

Although the temperature at which the pellets were exposed during drying (60 °C) is lower than the boiling point of water, the verified moisture loss is consistent with the thermogravimetric tests conducted with the clay mineral, montmorillonite, the main component of bentonites, which demonstrates that the thermal transitions associated with the elimination of adsorbed water can occur at temperatures below 100 °C [29,30].

It was also possible to observe that, both for the disintegration test and for the test in which there was static immersion in diesel and olefin, the values of lost mass (Tables 2 and 4) were lower than the values of mass reduction due to the loss of moisture presented in Table 3. This behavior is probably related to the disposition of organic compounds on the surfaces of the pellets, even in small amounts, increasing the mass registered after immersion, as shown in Table 4. This result demonstrated that, although the affinity between the organic compounds and the pellets is not expressive, this contact might form a physical barrier that is capable of hindering, even slightly, the elimination of water molecules during drying.

For the pellets that were immersed in glycerin, the absorption of this fluid was observed as a result of the interaction during the first two hours of the test in static condition, resulting in a mass increase from 20 to 26.06 g. This increase in mass is expressively significant, which represents a percentage of about 30.30% in relation to the mass of the dry sample, (Table 4) and it is attributed to the strong intermolecular interactions between glycerin, which has a hydrophilic nature, and the surfaces of the clay mineral [26].

This behavior, combined with the high boiling temperature of glycerin, which is 290 °C, inhibited the elimination of organic molecules adsorbed under the test conditions. Thus, the increase of approximately 17.60% in the mass that was verified after drying corresponds to the compensation between the absorption of organic molecules, which remain in the system and are responsible for the moistened appearance of the sample, and the elimination of water molecules, which corresponds to 13.90% (Table 3). Thus, the physical disintegration value of 14.36%, presented in Table 1, must be corrected by adding the mass increase resulting from the interaction between the pellets and glycerin (17.60%) so that the actual physical disintegration rate obtained is approximately 32%.

Table 5 presents the corrected values for the physical disintegration of the bentonite pellets in organic compounds, exclusively considering the loss of structural integrity provided by contact with these fluids. These results are based on observations during the performance of the test and on the interpretation of the results obtained for the additional tests that were performed while targeting the calculation of the moisture contents of the pellets.

**Table 5.** Corrected physical disintegration rates of pellets.

| Fluids | Corrected Disintegration Rates (%) |
|---|---|
| Diesel | −0.60 |
| Olefin | 0.24 |
| Glycerin | 31.96 |

For diesel and olefin, the corrected results showed that there was no relevant quantitative disintegration of the pellets, which was demonstrated by the visual appearances of these fluids, in which no disintegrated particles were observed. Furthermore, the negative value obtained for the disintegration rate of the pellets in diesel suggests that the mass of the sample immersed in this fluid, after drying, was greater than the initial mass of the sample without immersion, also after drying. Once there is no chemical interaction between the pellets and the diesel, it is inferred, therefore, that only the physical adsorption of the medium to the surfaces of the pellets occurs, and this mass increase is attributed to the thin oily layer adhered to the surfaces of the pellets. For glycerin, in turn, an even higher disintegration rate was observed after correction, which was attributed to the strong intermolecular interactions between glycerin and the pellet surface [26].

*3.3. Linear Swelling*

The percentage of linear swelling measured for the pellets, which were initially immersed in the organic compounds for 1 h and then immersed in deionized water for 24 h, is shown in the graph in Figure 10.

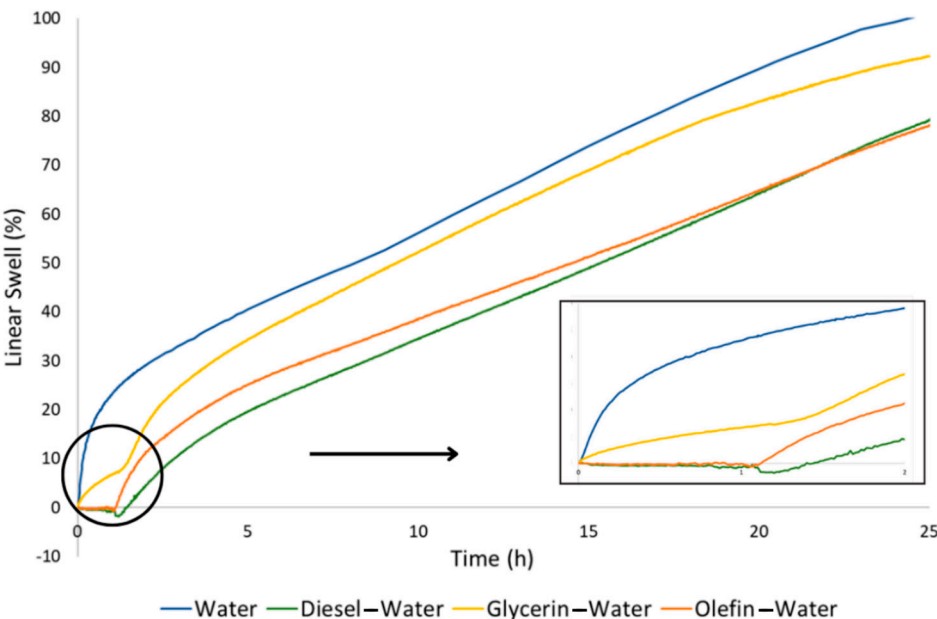

**Figure 10.** Percentage of linear swelling of bentonite pellets as a function of contact time in each organic compound (diesel, olefin and glycerin) followed by contact with deionized water.

During the first hour of the test, the bentonite pellets were in contact with the organic media, and different behaviors were observed. The results related to this time interval are better visualized in the amplified curves presented in the graph. The samples that were immersed in diesel and olefin did not show swelling, while the sample that was immersed in glycerin showed swelling from the beginning of the test, accounting for a total swelling close to 7% in 1 h.

As discussed previously, the interaction between organic compounds and bentonite might be attributed to the chemical properties of these substances. Glycerol, the main component of glycerin, has a hydrophilic nature [31], and when in contact with a mineral that is also hydrophilic, such as bentonite clay, it develops interactions with the surfaces of the particles that are electrically active [32], promoting an increase in the basal interplanar distance and, consequently, its swelling. In contrast, diesel and olefin are hydrophobic substances [33] and therefore do not develop interactions when in contact with a hydrophilic material [34].

After replacing diesel, olefin and glycerin with deionized water, a similar behavior was observed, regardless of the initial organic fluid: continuous swelling was recorded until the end of the 24 h of testing. For the tests that were carried out with the samples that were previously immersed in diesel and olefin, linear swelling values of approximately 78% were obtained. For the test carried out with the sample that was previously immersed in glycerin, the linear swelling obtained was 92%. Comparing these values, the linear swelling was 14% higher when glycerin was previously used. As already discussed, this behavior is most likely due to interactions between glycerol and clay particles that promoted an initial expansion between the clay layers. This first interaction caused the layers to be more loosely bound, favoring the hydration mechanisms as well as the hydration rates of the clay particles.

The linear swelling obtained for the samples that had previous contact with diesel and olefin evidenced that the immersion of the samples for a period of 1 h promoted the formation of a membrane on the surface of the tablets, and this, in turn, prevented the total hydration and swelling capacity of the samples. This was evidenced by the lower total linear swelling of the samples that were previously immersed in diesel and olefin, which was approximately 78%, while the swelling of the pellets that were only immersed in water (blue curve in the graph in Figure 10) was 101% at the end of 25 h.

Although the linear swelling test, carried out using the LSM equipment, registers the percentage of swelling of the sample vertically, it is possible to observe the swelling of the material radially through the metallic screen in which the sample is contained. This swelling was recorded and is shown in the images in Figure 11.

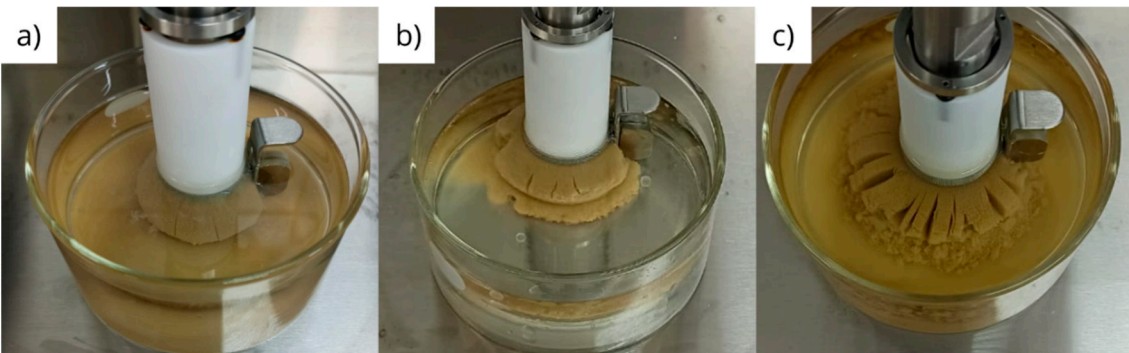

**Figure 11.** Visual appearances of bentonite pellets tablets after 24 h of hydration for samples previously immersed for 1 h in (**a**) diesel, (**b**) olefin and (**c**) glycerin.

Figure 11a,b exhibit a similarity in the radial swelling of the tablets that were previously immersed in diesel and olefin. The material has a uniform and cohesive appearance, although small openings are observed. For the tablets that were previously immersed in glycerin (Figure 11c), many openings are observed, which are larger than those noted in

the tablets that were immersed in diesel and olefin, suggesting the absence of cohesion between the particles.

The visual inspection of the pellets after 25 h of testing in the LSM, together with the other tests carried out and the analysis of the results, showed that the contact of bentonite pellets with glycerin favors the dispersion of the clay particles, compromising its physical integrity. On the other hand, diesel and olefin do not interact with the clay particles, do not interfere in the physical integrity of the pellets and, finally, do not affect their swelling, suggesting that these are promising fluids to be used for the displacement of bentonite pellets in offshore wells.

## 4. Conclusions

In this study, the compatibility of fluid media, consisting of saline solutions and organic compounds, with bentonite clay pellets was analyzed, aiming for its application as displacement fluid in offshore oil well abandonment operations. Based on the results obtained, the following were concluded:

1.  Saline solutions consisting of sodium chloride (NaCl), calcium chloride ($CaCl_2$) and potassium chloride (KCl), in different saturations, compromise the physical integrity of bentonite pellets and are unsuitable for use as displacement fluids.
2.  Saline solutions with lower saturations, around 10 to 20%, promote faster and more significant disintegration of bentonite pellets.
3.  The interaction between glycerin and bentonite pellets results in a partial physical disintegration of the pellets, which is significantly enhanced by shear under dynamic conditions.
4.  Bentonite pellets maintain their physical integrity when exposed to diesel and olefin, even under dynamic conditions.
5.  Diesel and olefin seem to present minor effects on pellet hydration and swelling, since linear swelling of 78% in deionized water was observed after the previous contact with these fluids, which represents a decrease of only 20% in its total swelling capacity in water. It does not impair the formation of a hydraulically solid plugs of bentonite in the well, and neither does the use of diesel and olefin as displacement fluids.
6.  Among the fluids considered for the displacement of pellets in offshore wells, diesel and olefin proved to be highly promising alternatives.

**Author Contributions:** Conceptualization, W.R.P.d.C. and L.V.A.; methodology, L.R.C.d.O., M.C.d.S.L. and A.C.A.C.; validation, T.A.d.O.; formal analysis, M.B., investigation, W.R.P.d.C., K.C.N. and R.L.d.N.G.; resources, E.A.d.S.; writing—original draft preparation, W.R.P.d.C., M.C.d.S.L. and L.R.C.d.O.; writing—review and editing, L.V.A. and R.C.A.d.M.N.; visualization, W.R.P.d.C.; supervision, K.C.N.; project administration, L.V.A.; funding acquisition, R.C.A.d.M.N. and E.A.d.S. All authors have read and agreed to the published version of the manuscript.

**Funding:** This research was funded by Petrobras (Petróleo Brasileiro S.A.), grant number 0050.0120134.21.9.

**Data Availability Statement:** Data are contained within the article.

**Conflicts of Interest:** The authors declare no conflict of interest.

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
