# Peer review of "Evaluation of Saline Solutions and Organic Compounds as Displacement Fluids of Bentonite Pellets for Application in Abandonment of Offshore Wells"

_processes, doi:10.3390/pr11123375_

Round 1

Reviewer 1 Report

Comments and Suggestions for Authors

Inhibition or delaying bentonite pellet swelling by using oil-base drilling fluids has been extensively studied, the same fundamental aspect must be applied to understand bentonite pellet swelling. For such reason, I see the results of swelling of bentonite pellets are similar to those obtained by using oil-base drilling fluids to stabilize clays and drill clay-rich zones. 

Or could you clearly explain why the behavior of the pellet could be different?

Reviewer 2 Report

Comments and Suggestions for Authors

The manuscript “Evaluation of saline solutions and organic compounds as displacement fluids of bentonite pellets for application on abandonment of offshore wells” aims to test different displacement fluids for seal off with bentonite pellets the abandonment offshore wells. The contribution of this paper is of interest in the scientific community. Prior to publication, this manuscript need some improvement:

Introduction: Give an overview of the main features of the bentonite, such as mineralogy, chemistry, fisico-mechanical properties, focusing on hydrophilic behaviour that is the aim of this experimental study. This paragraph should include a description of the displacement fluids, illustrating pro and cons of the main used in abandonment wells.

Materials and methods: this paragraph should specify the origin of bentonite used in the experiment, if it is natural or artificial, and its provenance and should include some information reported in the Results.

L93: instead of listing all percentage in saturation, explain in different way that the brines were prepared in increment of 10% from a concentration of 10% to 100%.

Please also specify in a table how many mg/L each percentage of salt corresponds to.

L99: it would be better for each test to be performed on 3 replicates.

L108: please specify the main requirements of the ISO 10416:2008

L112: please use the extended form of the abbreviation at the first use, such as for ABNT number and psi in L134

L123: describe as the dispersibility or disintegration rate is calculated

Results need more details for ensuring the reproducibility of the experiments.

L142: which data analysis were performed with the software LSM 2100? Statistics? Please specify

L151-L154 these sentences should be moved to material & methods. Also references are needed.

L156: improve the readability avoiding to finish and starting sentences with Fig. Maybe use the extended form such as “fluid, is shown in Fig. 2. In figure 3 and 4 are reported the appearance observed at the end of the test”

L162: specify the number with letter in the figure (eg 5s, 4s, 9s)

L167:  the sentence starts with “as seen …” but actually the pictures are not so clear. I suggest to describe in the text the main features that the pictures shown, which changes they observed, and they should add a picture of the pellet without treatment as test control

L183: Zhang et al. (2016) reference numeber is missing

L187: describe destabilization of bentonite particles

L192-L196: the meaning of this sentence is not clear, please rephrased it

L217-L220: the meaning of this sentence is not clear. Interaction between glycerol and what? please rephrased

L235: how the authors evaluated the pellet cohesion? Just visually? This aspect should be discussed in material& methods

L262: Which values are expected? Quantify please

L266: in material & methods should be explained as the average mass reduction by loss of moisture in the pellets is calculated

L268-L271: this sentence should be moved in the materials & methods

L274-L275: data are the mean of different measurements or are from a single analysis?

L286: “As this amount of water removed does not compromise…” invert water and removed as removed water

L290: “the moisture loss verified is consistent…” use the verified moisture loss

L304-L305: “condition, resulting in a mass of 26.06g.” it is not clear if the mass increase from 20g to 26.06 g or there was an increasing of 26 g.

L305: “This mass increase is expressively significant, which…” use This increase in mass is

L315: how the physical disintegration value must be corrected? Describe

L344: describe the small sketch in the fig 10. Add also 1-hour interval in the X-axis for the first 5 hours, so that the first hour can be recognised

L377: As the authors note, the swelling is 3dimensional so the increased volume should be assessed

 Reference: Check the form of reference n. 3, 3 30

Comments on the Quality of English Language

Moderate editing of English language is required 

Reviewer 3 Report

Comments and Suggestions for Authors

Dear authors,

This manuscript is suitable for this journal.

The research aim is also important for solving the problems of offshore oil well abandonment operations.

It has been shown that bentonite plug is another good alternative method for cement plug. few studies 80

analyze the operational performance prior to the swelling stage, which is crucial, since it is related to the interaction between the pellets and the fluids that must provide their hydraulic displacement efficiently.

Generally, there is fine for abstract and introduction.

For the methods and results, there are missing controls. the control should also write down and show in the results.

The results indicate the targeting its application as displacement fluid in abandonment operations of offshore oil wells according their tests.

Although some marine organisms may affect the operations, but this is another further research.

Round 2

Reviewer 1 Report

Comments and Suggestions for Authors

NaCl in table 1 must be corrected.

Author Response

Reviewer comment: NaCl in table 1 must be corrected.

Response: The correction was done and is highlighted in green.